# Ectopic Over-Expression of *BjuAGL9-2* Promotes Flowering and Pale-Yellow Phenotype in *Arabidopsis*

**DOI:** 10.3390/plants14223502

**Published:** 2025-11-17

**Authors:** Guoqiang Han, Keran Ren, Rongyan He, Ruirui Mo, Jing Zeng, Mingming Sui

**Affiliations:** 1College of Rural Revitalization, Fuyang Institute of Technology, Fuyang 236031, China; hgq-23@163.com; 2College of Agronomy and Biotechnology, Southwest University, Chongqing 400716, China; alkaid_ren@163.com; 3School of Life Advanced Agriculture Bioengineering, Yangtze Normal University, Chongqing 408100, China; 19534252733@163.com (R.H.); morr056@163.com (R.M.)

**Keywords:** *B. juncea*, interaction proteins, over-expression lines, *BjuAGL9-2*, early flowering, pale-yellow

## Abstract

*Brassica juncea* is an important leafy vegetable, and flowering time is a key determinant of its yield and quality. In this study, one significantly up-regulated gene, *BjuAGL9-2*, was identified from RNA-Seq data. qRT-PCR analysis confirmed that *BjuAGL9-2* expression was significantly elevated in reproductive organs and reproductive stages. Further five *BjuAGL9-2* over-expression (OE) lines were subsequently generated, which showed an early-flowering and pale-yellow leaf phenotype compared to the wild type. qRT-PCR assays found that the mRNA of core floral integrator genes was changed in *Arabidopsis* OE lines. Yeast two-hybrid (Y2H) and bimolecular fluorescence complementation (BiFC) assays indicated that BjuAGL9-2 interacted with BjuTUA5, BjuZFP7, BjuGSTU5, and BjuMAPK16 in vivo. Sub-cellular localization assays showed that BjuAGL9-2 localizes in the nucleus, whereas its interacting partners localize in the cytoplasm. qRT-PCR assays further revealed that *BjuTUA5* and *BjuGSTU5* were up-regulated in flower buds, while *BjuZFP7* and *BjuMAPK16* were down-regulated. During vegetative stages, all four genes were up-regulated in *B. juncea*. As for BjuAGL9-2 interaction protein-encoding homolog genes, except *AtGSTU5*, the other three genes were up-regulated in *Arabidopsis* OE lines. Additionally, qRT-PCR analysis of chlorophyll biosynthesis-related genes showed that 19 of 27 genes were up-regulated, while 8 genes were down-regulated, in *Arabidopsis* OE lines. Collectively, these findings suggest that *BjuAGL9-2* promotes flowering and contributes to the pale-yellow phenotype by regulating its interacting protein-coding genes, floral integrators, and chlorophyll biosynthesis genes.

## 1. Introduction

*Brassica juncea* (L.) Czern. is an important leafy cruciferous vegetable that is widely consumed in China, particularly as a deeply processed pickle. As a vegetative-harvested crop, delaying flowering to increase yield and quality is a major breeding objective in *B. juncea*. In *Arabidopsis thatliana*, the flowering time regulatory network has been extensively studied. Six major signaling pathways, the autonomous, photoperiod, vernalization, gibberellic acid, ambient temperature, and aging pathways, have been reported to play critical roles in flowering regulation [1]. Within these pathways, multiple transcription factor families, including bZIP, MADS-box, WRKY, and MYB participate in control of flowering [2,3]. As plants transition to the reproductive stage, signals from these six pathways converge on floral integrator genes *SOC1* (*suppressor of over-expression of CO1*) and *FT* (*flowering locus T*). Then *SOC1* and *FT* further transmit these signals to floral meristem identification genes *LFY* (*leafy*), *AP1* (*apetala 1*), and *FUL* (*fruitfull*), thereby initiating flowering [1,4,5,6,7].

The MADS-box transcription factor family has been widely reported to participate in diverse processes of plant growth and development, particularly in inflorescence architecture and floral organ formation [8,9]. *Arabidopsis thaliana* contains 108 MADS-box members, while *Brassica rapa*. *Zea mays*, and *Vitis vinifera* harbor 160, 87, and 54 members, respectively [10,11,12,13]. Based on sequence characteristics, the MADS-box family can be classified into two major types. Type I genes typically contain a conserved MADS-box domain together with a highly *C*-terminal region [14]. Type II can be subdivided into MIKC^c^-type and MIKC*-type clade [15,16]. MICK-type genes usually harbor four conserved domains. The MADS-box domain functions as a DNA-binding region, recognizing and binding to the CArG-box motif [17,18]. The I (intervening) domain specifically binds to downstream targets [19]. The K (keratin-like) domain facilitates formation of higher-order MADS-box protein complexes [20,21]. However, the function of the C-terminal domain remains unclear.

ABCDE is a well-known model to explain floral organ determination at the genetic level [22]. In *Arabidopsis*, except *Apetala2*, all other genes belong to the MADS-box family [23]. *AtAP1* (*apetala1*) is a member of the A-class genes and is required for floral meristem development and floral organ identity [24]. The B-class genes *AtAP3* (*apetala3*) and *PI* (*pistillata*) control petal and stamen identity [25]. The C-class gene *AG* (*agamous*) is essential for reproductive organ identity and floral meristem determinacy [26]. The D-class genes *STK* (*seedstick*), *SHP1* (*shatterproof1*), and *SHP2* (*shatterproof2*) regulate ovule identity [27,28]. The E-class genes *SEP1* (*AGL2*, *sepallata1*), *SEP2* (*AGL4*, *sepallata2*), *SEP3* (*AGL9*, *sepallata3*), and *SEP4* (*sepallata4*) are required for sepal, petal, stamen, and carpel development [29,30,31,32,33].

In this study, *BjuAGL9-2* was found to be significantly up-regulated in flower organs and reproductive stages. Transgenic over-expression *BjuAGL9-2 Arabidopsis* showed an early-flowering and pale-yellow phenotype. qRT-PCR assays found that the mRNA expression levels of *AtFT*, *AtSOC1*, *AtCO* (*constans*), *AtFUL*, *AtSVP*, and *AtAP1* were down-regulated, but *AtLFY* and *AtLC* were up-regulated, in *Arabidopsis* OE lines compared to the wild type. Y2H and BiFC assays confirmed that BjuAGL9-2 interact with BjuTUA5, BjuZFP7, BjuGSTU5, and BjuMAPK16 in the nucleus. Sub-cellular location assays indicated that BjuAGL9-2 localized the nucleus, whereas its interacting partners were localized the cytoplasm. qRT-PCR analysis showed that *BjuTUA5* and *BjuGSTU5* were up-regulated in flower buds, while *BjuZFP7* and *BjuMAPK16* were down-regulated. During vegetative stages, all four genes were up-regulated in *B. juncea*. And in *Arabidopsis* OE lines, except *AtGSTU5*, the other three homologs of BjuAGL9-2 interaction protein-encoding genes, *AtTUA5*, *AtZFP7*, and *AtMAPK16*, were all up-regulated compared to the wild type. Further, qRT-PCR assays revealed that among 27 chlorophyll biosynthesis-related genes, 19 genes were up-regulated and 8 genes were down-regulated in *Arabidopsis* OE lines. Collectively, these findings suggest that *BjuAGL9-2* promotes flowering and contributes to the pale-yellow phenotype by regulating the expression of its interacting protein-coding genes, floral integrators, and chlorophyll biosynthesis genes.

## 2. Results

### 2.1. BjuAGL9-2 Functions in Promoting Plant Flowering and Pale-Yellow Phenotype in A. thaliana

To compare gene expression profiles in *B. juncea* shoot tips at vegetative and flowering stages, RNA-Seq was recruited to analysis differentially expressed genes (DEGs). One gene named *BjuAGL9-2* was found to be significantly up-regulated at the reproductive stage. qRT-PCR assays further confirmed that *BjuAGL9-2* was up-regulated in petioles, leaves, flower buds, and flowers, with particularly strong expression in flower buds and flowers compared to roots (Figure 1a). Across different developmental stages, *BjuAGL9-2* expression was elevated in reproductive stages, showing significant up-regulation at both squaring and bolting stages compared to the vegetative stage (Figure 1b).

**Figure 1 plants-14-03502-f001:**
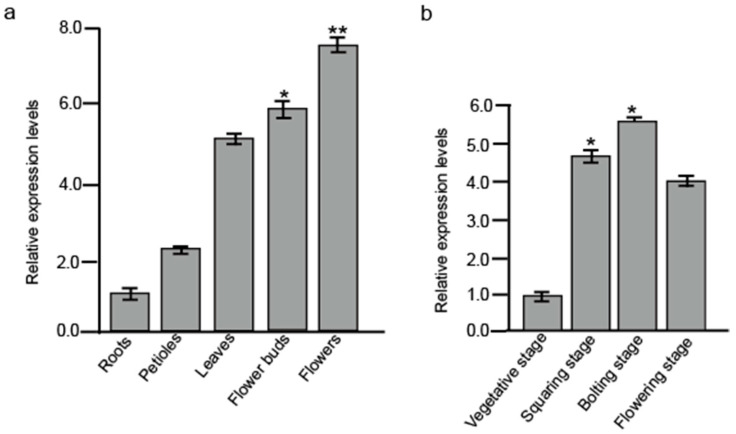
*BjuAGL9-2* expression levels in different tissues and at different development stages. (**a**) *BjuAGL9-2* expression levels in different tissues. (**b**) *BjuAGL9-2* expression levels at different development stages. * *p* < 0.05, ** *p* < 0.01.

To investigate the function of *BjuAGL9-2* in plant growth and development, *Arabidopsis* OE plants were generated (Figure 2). qRT-PCR analysis indicated that *BjuAGL9-2* transcript levels were elevated in all five OE lines (Figure 2c). Among them, three lines (OE3, OE4, and OE5) displayed strong induction, with approximately 10-fold, 20-fold, and 15-fold higher expression compared to the wild type, respectively (Figure 2c). Western blot analysis further confirmed the accumulation of GFP-BjuAGL9-2 protein in all five OE lines (Figure 2d). Phenotype observations reveal that 25 of OE1, 26 of OE2, 27 of OE3, 28 of OE4, and 27 of OE5 plants initiated flowering approximately 28 days after transfer to nutrient soil, whereas the wild type flowered at around 35 days. qRT-PCR assays indicated that the mRNA expression levels of *AtFT*, *AtSOC1*, *AtCO*, *AtFUL*, *AtSVP*, and *AtAP1* were down-regulated, in which *AtFT*, *AtSOC1*, *AtCO*, *AtFUL*, and *AtAP1* were significantly down-regulated, but the mRNA expression levels of *AtLFY* and *AtLC* were significantly up-regulated in *Arabidopsis* OE lines compared to the wild type (Figure 2e,f). In addition, all five transgenic lines showed a pale-yellow phenotype from the four-true-leaf stage to the flowering stage (Figure 2). These results indicate that the over-expression of *BjuAGL9-2* promotes flowering and induces the pale-yellow phenotype.

### 2.2. BjuAGL9-2 Interaction Protein Screening

To detect the protein interaction network in regulating plant flowering, the bait vector pGBKT7-BjuAGL9-2 was used to screen potential candidates from a *B. juncea* bolting stage flower buds yeast library, which was constructed by using the Gateway cloning system. The length of PCR amplification fragments is approximately 900 bp, 800 bp, 800 bp, 800 bp, 800 bp, 800 bp, 500 bp, 800 bp, 800 bp, 800 bp, 500 bp, and 800 bp, respectively (Appendix A). Sequencing analysis identified several putative BjuAGL9-2-interacting proteins, including BjuANXAD1, BjuTUA5, BjuZFP7, BjuGSTU5, BjuVA09G18110, BjuWRKY11, BjuRHY1A, BjuVA01G03870, and BjuMAPK16 (Table 1).

### 2.3. Interactive Analyses Between BjuAGL9-2 and Screened Proteins

To confirm the interaction between BjuAGL9-2 and candidate proteins, which were identified from the yeast library, pGBKT7-BjuAGL9-2 and each of the prey constructs (pGADT7-BjuANXAD1, pGADT7-BjuTUA5, pGADT7-BjuZFP7, pGADT7-BjuGSTU5, pGADT7-BjuVA09G18110, pGADT7-BjuWRKY11, pGADT7-BjuRHY1A, pGADT7-BjuVA01G03870, and pGADT7-BjuMAPK 16) were co-transformed into Gold strain, pairwise. Growth and reporter activation on QDO medium demonstrated that BjuAGL9-2 interacts with BjuTUA5, BjuZFP7, BjuGSTU5, and BjuMAPK16 in vivo (Figure 3).

To further confirm the interaction between BjuAGL9-2 with its candidate partners (BjuTUA5, BjuZFP7, BjuGSTU5, and BjuMAPK16), BiFC assays were performed in *Nicotiana benthamiana* leaf cells. Split YFP complementation assays revealed strong yellow fluorescent signals in the nucleus when BjuAGL9-2 was co-infiltrated with each of the candidate proteins (Figure 4). These results confirmed that BjuAGL9-2 interacted with BjuTUA5, BjuZFP7, BjuGSTU5, and BjuMAPK16 in the nucleus (Figure 4).

### 2.4. Sub-Cellular Location Assays

To analyze the sub-cellular location of BjuAGL9-2, BjuTUA5, BjuZFP7, BjuGSTU5, and BjuMAPK16, their coding sequences were fused to *GFP* and transiently expressed in *Nicotiana benthamiana* epidermal cells. Fluorescence microscopy revealed that GFP-BjuAGL9-2 localized predominantly to the nucleus, whereas GFP-BjuTUA5, GFP-BjuZFP7, GFP-BjuGSTU5, GFP-BjuMAPK16, and GFP localized to the cytoplasm (Figure 5).

### 2.5. Interactive Protein-Encoding Gene Expression Analysis in Different Tissues and at Different Development Stages in B. juncea

To examine the mRNA expression levels of genes encoding BjuAGL9-2-interacting proteins (*BjuTUA5*, *BjuZFP7*, *BjuGSTU5*, and *BjuMAPK16*) in different tissues, qRT-PCR analysis was conducted (Figure 6a). *BjuTUA5* was up-regulated in petioles, leaves, flower buds, and flowers, with particularly strong expression in reproductive organs (flower buds and flowers), compared to roots. *BjuZFP7* showed elevated expression in petioles, leaves, and flowers, with significant up-regulation in petioles and leaves but was significantly down-regulated in flower buds compared to roots. *BjuGSTU5* displayed about 2-fold changes in petioles, leaves, and flowers but was specifically and significantly up-regulated in flower buds. *BjuMAPK16* was up-regulated in petioles, leaves, and flowers, with a significant increase in leaves and flowers, but was down-regulated in flower buds compared to roots (Figure 6a).

Analysis of mRNA expression at different developmental stages revealed that *BjuTUA5*, *BjuGSTU5,* and *BjuMAPK16* were generally up-regulated during reproductive stages. In particular, *BjuTUA5* exhibited significant changes in expression across all reproductive stages, while *BjuMAPK16* was markedly up-regulated at both the squaring and flowering stages (Figure 6b). In contrast, *BjuZFP7* showed increased expression at both the squaring and flowering stages but was down-regulated at the bolting stage (Figure 6b). Regarding the impact of *BjuAGL9-2* on the expression levels of homologs of interaction protein-encoding genes, qRT-PCR assays indicated that whereas *AtGSTU5* was down-regulated, *AtTUA5*, *AtZFP7*, and *AtMAPK16* were up-regulated in *Arabidopsis* OE lines compared to the wild type (Figure 6c).

### 2.6. Chlorophyll Synthesis Gene Expression Levels in Arabidopsis OE Lines

To investigate the effect of *BjuAGL9-2* on chlorophyll biosynthesis, qRT-PCR assays were performed on wild-type and *Arabidopsis* OE lines (Appendix A). For the first step of chlorophyll biosynthesis, all genes were up-regulated in OE lines except *AtHEMA1*, which was down-regulated. Specifically, *AtHEMA3*, and *AtHEML* were up-regulated compared to the wild type (Figure 7a). For the second step, *AtHEMB1*, *AtHEMF1*, and *AtHEMGs* were down-regulated, whereas *AtHEMB2*, *AtHEMC*, *AtHEMD*, *AtHEMEs*, and *AtHEMF2* were up-regulated, in OE lines (Figure 7b). For the third step, multiple genes also showed altered expression. *AtCHLs*, *AtCRD1*, *AtDVR*, *AtPORs*, and *AtCHLG* were up-regulated, while *AtCHIs* and *AtCAO* were down-regulated, in *Arabidopsis* OE lines compared to the wild type (Figure 7c).

## 3. Discussion

In plants, flowering time is regulated by six major pathways, in which *FT* and *FLC* act as key integrators. In this study, *BjuAGL9-2*, a member of the MADS-box family, was identified as being up-regulated in reproductive organs and reproductive stages (Figure 1). To investigate *BjuAGL9-2*’s role in plant growth, *BjuAGL9-2* OE lines were generated (Figure 2). Three *Arabidopsis* OE lines with significantly elevated expression levels of *BjuAGL9-2* showed early-flowering compared to the wild type (Figure 2). As plant floral integrator genes, *FT*, *SOC1*, *CO*, *FUL*, *AP1*, and *LFY* function in promoting flowering, but *FLC* and *SVP* function in delaying flowering [1,4,5,6,7]. qRT-PCR assays indicated that the floral integrator genes *FT*, *SOC1*, *CO*, *FUL*, and *AP1* were down-regulated, but *LFY* and *FLC* were significantly up-regulated, in *Arabidopsis* OE lines. This means that *BjuAGL9-2* functions in regulates both positive and negative floral integrator genes. To further elucidate the molecular regulatory network of *BjuAGL9-2* in flowering, Y2H and BiFC assays were performed. The results demonstrated that BjuTUA5, BjuZFP7, BjuGSTU5, and BjuMAPK16 interact with BjuAGL9-2 in the nucleus (Table 1, Figure 3 and Figure 4). Sub-cellular location assays confirmed that BjuAGL9-2 is localized in the nucleus, whereas its interacting proteins are localized in the cytoplasm (Figure 5). This means that the interacting proteins may be transported to nucleus for their interaction and function in prompt flowering.

*BjuVA10G20230* was identified as *TUA5* (*tubulin alpha-5 chain*), suggesting functions in pollen development, while *AGL9* is highly expressed in stamens, indicating its involvement in stamens development [32,34,35]. In this study, qRT-PCR analysis showed that *BjuTUA5* was up-regulated in reproductive organs (flower buds and flowers) and at reproductive stages, including the squaring and bolting stages (flower buds) and the flowering stage (flowers, Figure 6). In *Arabidopsis* OE lines, its homolog gene *AtTUA5* was up-regulated compared to the wild type (Figure 6c). These findings suggest that *AGL9-2* may function in stamen development by regulating *AtTUA5* at both the transcription and translation levels. *BjuVA08G27100* was identified as *ZFP7* (*zinc finger protein 7*), which has been reported to be down-regulated by 90% in floral organs from stages 4 to 12 [36]. Consistent with these findings, in this study, *BjuZFP7* expression was down-regulated in flower buds and at reproductive stages in *B. juncea* (Figure 6), further supporting its role in floral organ development. However, in *Arabidopsis* OE lines, the expression level of *AtZFP7* was slightly up-regulated (Figure 6c). This means that *BjuAGL9-2* functions by negatively regulating *BjuZFP7* expression in flower organs but positively regulates *BjuZFP7* expression in leaves. *BjuVA03G025670* was identified as *GSTU5* (*glutathione S-transferase*) and has been implicated in defense response [37]. qRT-PCR assays revealed that *BjuGSTU5* was significantly up-regulated in flower buds and reproductive stages (squaring, bolting, and flowering stages, Figure 6). Similarly, *BjuVA10G20860* was identified as *MAPK16* (*mitogen-activated protein kinase 16*), which is also involved in defense response [38]. Its expression was significantly up-regulated in flowers and reproductive stages (squaring, bolting, and flowering, Figure 6). However, in *Arabidopsis* OE lines, the homologous gene *AtGSTU5* was down-regulated, but *AtMAPK16* was up-regulated compared to the wild type (Figure 6c). These results suggest that the interaction between BjuAGL9-2 and BjuVA03G025670/MAPK16 may play an opposite role in defense-related processes during reproductive development.

Chlorophyll content largely determines plant color, and its biosynthesis can be categorized into three steps [39]. In this study, *Arabidopsis* OE lines displayed a pale-yellow phenotype (Figure 2a,b). To explore the molecular basis of this phenotype, qRT-PCR assays were conducted to examine the expression of chlorophyll biosynthesis genes. Among the five genes in the first step, *AtHEMA1* was down-regulated, while *AtHEMA2*, *AtHEMA3*, *AtHEML1*, and *AtHEML2* were up-regulated in *Arabidopsis* OE lines compared to the wild type (Figure 7a). Previous studies have shown that *HEMA3* and *HEMA2* do not contribute to GluTR activity, whereas *HEMA1* is the key enzyme for ALA synthesis in plants [40,41]. The down-regulation of *AtHEMA1* observed in *Arabidopsis* OE lines is consistent with their pale-yellow phenotype (Figure 7a). Furthermore, *HEML2* has been identified as a positive regulator of chlorophyll biosynthesis, with *heml2* mutants showing reduced chlorophyll a/b content, while *heml1* mutants do not display such a decrease [42]. In this study, the up-regulation of *AtHEML1* and *AtHEML2* in *Arabidopsis* OE lines suggests that *BjuAGL9-2* may influence chlorophyll a/b accumulation through the induction of *AtHEML1* expression, even though the repression of *AtHEMA1* ultimately contributes to the pale-yellow phenotype.

*HEMB1* and *HEMB2* are involved in the second step of chlorophyll biosynthesis and have been reported to function in plant immunity and the heme biosynthetic pathway, respectively [43,44]. In *Arabidopsis* OE lines, *AtHEMB1* and *AtHEMB2* were slightly up-regulated, compared to the wild type (Figure 7b), suggesting that *BjuAGL9-2* may positively regulate *AtHEMB* expression. *HEMC*, encoding porphobilinogen deaminase (PBGD), and *HEMD*, encoding uroporphyrinogen III synthase (UROS), both function in the early stages of chlorophyll and heme biosynthesis [45,46]. *HEME* genes, encoding UROD, catalyze the decarboxylation of uroporphyrinogen III to coproporphyrinogen III [47,48]. In this study, *AtHEMC*, *AtHEMD*, *AtHEME1*, and *AtHEME2* were all up-regulated in *Arabidopsis* OE lines compared to the wild type (Figure 7b), indicating that *BjuAGL9-2* positively regulates these genes during early chlorophyll and heme biosynthesis. *HEMF* genes are involved in leaf development, and mutants exhibit developmentally regulated and light-dependent leaf lesions [49]. In this study, only *AtHEMF2* was significantly up-regulated, while *AtHEMF1* was slightly down-regulated, in *Arabidopsis* OE lines compared to the wild type (Figure 7b). These results suggest that *BjuAGL9-2* contributes to the yellowing phenotype primarily through the regulation of *AtHEMF2*. *HEMG* genes, encoding protoporphyrinogen oxidase, are associated with the lesion-mimic phenotype when mutated [50]. In this study, *AtHEMG* was slightly down-regulated, consistent with the absence of the lesion-mimic phenotype in *Arabidopsis* OE lines (Figure 7b). This indicates that *BjuAGL9-2* does not regulate *HEMG* genes.

The third step of chlorophyll biosynthesis involves 12 genes (*CHLH*, *CHLI1*, *CHLI2*, *CHLD*, *CHLM*, *CRD1*, *DVR*, *PORA*, *PORB*, *PORC*, *CHLG*, and *CAO*) and seven enzymes (MgCh, MgPMT, MgPME, DVR, POR, CHLG, and CAO) [39]. The MgCh enzyme consists of three subunits and is encoded by *CHLH*, *CHLI1*, *CHLI2*, and *CHLD*, respectively [51,52,53,54]. In *Arabidopsis* OE lines, *AtCHLH* and *AtCHLD* were up-regulated, while *AtCHLI1* and *AtCHLI2* were down-regulated, compared to the wild type (Figure 7c). In *Arabidopsis*, mutations in CHLI subunits result in a yellow-green phenotype [55]. Thus, the down-regulation of *AtCHLI1* and *AtCHLI2* in *Arabidopsis* OE lines is consistent with the observed pale-yellow phenotype (Figure 7c). *CHLM* functions as a positive regulator of chlorophyll–protein complex formation [56]]. In this study, *AtCHLM* was significantly up-regulated in *Arabidopsis* OE lines, indicating that *BjuAGL9-2* positively regulates chlorophyll–protein complexes (Figure 7c). *CRD1* is also involved in chlorophyll synthesis, and its mutation leads to an elevated chlorophyll a/b ratio and a pale-green phenotype [57]. In this study, *AtCRD1* was up-regulated in *Arabidopsis* OE lines (Figure 7c), suggesting that it may not contribute to the pale-yellow phenotype. *DVR* is required for chlorophyll synthesis, and loss-of-function mutants showed reduced chlorophyll contents and a pale-green phenotype [58]. In this study, *AtDVR* was only slightly up-regulated in *Arabidopsis* OE lines (Figure 7c), indicating that it does not contribute to the pale-yellow phenotype. Protochlorophyllide oxidoreductase (POR) is a key enzyme for chlorophyll synthesis that promotes the greening of etiolated plants. In *Arabidopsis*, there are three POR isoforms. The ectopic expression of *PORA* in the *porB-1 porC-1* double-mutant rescues growth defects and chlorophyll deficiency, demonstrating that *PORA* is an essential component for bulk chlorophyll biosynthesis [59,60,61],,. In *Arabidopsis* OE lines, all three genes (*AtPORA, AtPORB, and AtPORC)* were up-regulated, suggesting that they are not responsible for the pale-yellow phenotype in *Arabidopsis* OE lines (Figure 7c). *CHLG* functions as a positive regulator of ALA synthesis, and antisense suppression of *CHLG* reduces ALA production [62]. In our study, *AtCHLG* was up-regulated in *Arabidopsis* OE lines, indicating that *BjuAGL9-2* positively regulates *CHLG* expression (Figure 7c). Finally, *CAO* is required for chlorophyll b precursor biosynthesis [63]. qRT-PCR assays showed that *AtCAO* was down-regulated in OE lines (Figure 7c), suggesting that *BjuAGL9-2* influences the chlorophyll b content through regulation of *AtCAO*. All those results suggesting that *BjuAGL9-2* functions in the pale-yellow phenotype by regulating *AtHEMA1*, *AtHEML1*, *AtHEMF2*, *AtCHLI1*, *AtCHLI2*, and *AtCAO* in *Arabidopsis* OE lines.

Taken together, the results of this study demonstrate that *BjuAGL9-2* promotes flowering and induces a pale-yellow phenotype in *Arabidopsis*. A qRT-PCR assay confirmed that the floral integrator genes *FT*, *SOC1*, *CO*, *FUL*, and *AP1* were down-regulated, but *LFY* and *FLC* were significantly up-regulated, in *Arabidopsis* OE lines. BjuAGL9-2 is nuclear-localized, but its interacting proteins BjuTUA5, BjuZFP7, BjuGSTU5, and BjuMAPK16 reside in the cytoplasm. The mRNA expression levels of *BjuAGL9-2*, *BjuTUA5*, and *BjuGSTU5* were up-regulated in flower buds, whereas *BjuZFP7* and *BjuMAPK16* were down-regulated. Notably, all four interacting genes were up-regulated during vegetative stages in *B. juncea*. In *Arabidopsis* OE lines, while *AtGSTU5* was down-regulated, *AtTUA5*, *AtZFP7*, and *AtMAPK16* were up-regulated, compared to the wild type. In addition, expression profiling of chlorophyll biosynthesis genes showed that 19 genes were up-regulated and 8 genes were down-regulated in *Arabidopsis* OE lines, providing a molecular explanation for the pale-yellow phenotype. Collectively, these findings suggest that *BjuAGL9-2* integrates flowering and leaf color regulation into chlorophyll metabolism by modulating key transcriptional and protein interaction networks.

## 4. Materials and Methods

### 4.1. Plant Growth Conditions

*B. juncea* plants were grown under natural field conditions (18–23 °C; 8 h photoperiod/16 h dark). For tissue-specific expression analysis, total RNA was extracted from roots, petioles, leaves, flower buds, and flowers. For developmental stage-specific expression analysis, RNA was extracted from leaves at the vegetative stage (four-true-leaf stage), flower buds at the squaring (first flower buds observed with a diameter of 3 mm) and bolting stages (before the first flower was observed), and flowers (the first flower observed) at the flowering stage. Total RNA was extracted using the RNAprep Pure Plant Kit (Tiangen, Beijing, China), and 1 μg of total RNA was used for cDNA synthesis (Appendix A). The ALL-In One 5*RT MasterMix (abm, Zhenjiang, China) kit was used for cDNA synthesis, and the synthesis program was 37 °C, 15 min, 60 °C, 10 min, 95 °C 3 min.

### 4.2. BjuAGL9-2 Over-Expression Line Screening and Phenotype Observation

The coding sequence of *BjuAGL9-2* was cloned into the p1300-GFP vector under the control of the CaMV35S promoter. The transformation of *Arabidopsis thaliana* was performed using the *Agrobacterium*-mediated (GV3101 strain with a pJIC SA_Rep plasmid) floral dip method [64]. T_0_ transgenic plants were initially screened on hygromycin-containing medium, and genomic DNA from leaves was used to identify 21 lines of T_1_ generation, 15 lines of T_2_ generation, and 8 lines of T_3_ generation. For gene expression analysis, total RNA was isolated from leaves of *Arabidopsis* homozygous OE lines 3, 4, and 5, as well as wild-type plants. For phenotype observation, 30 seeds of each *BjuAGL9-2* OE lines were sown on MS solid medium, left at 4 °C for 3 d, transferred to 23 °C under continuous white light (16 h light/8 h dark) until the growth of four true leaves, and then transferred to nutrient soil. Transcripts of *BjuAGL9-2*, the interaction protein-encoding genes, chlorophyll biosynthesis-related genes, the homologs of interacting protein-encoding genes (*AtTUA5*, *AtZFP7*, *AtGSTU5*, and *AtMAPK16*), and floral integrator genes were amplified using specific primers (Appendix A). qRT-PCR was performed on the Roche LightCycler 480II (Roche, Basel, Switzerland), with 40 cycles, with *Atactin* and *Bjuactin2* serving as the internal control. The reaction mixture was 2× qPCR mix 5 μL, cDNA 4.2 μL, primer-F: 0.8 μL, primer-R: 0.8 μL. And the PCR program was as follows: 40 cycles of 95 °C for 30 s; 95 °C for 10 s; 58 °C for 10 s; 72 °C for 10 s. Relative gene expression was calculated using the 2^−ΔΔCt^ method, and each sample was analyzed in triplicate for both biological and technical replicates. For protein detection, total protein was extracted from transgenic plants using extraction buffer (50 mM Tris-HCl, pH 7.5; 100 mM NaCl; 1 mM EDTA, pH 8.0; 10% glycerol; 0.5% SDS). Western blot assays were performed using anti-GFP antibody (ABMART, Shanghai, China), with total protein from the *A. thaliana* wild type serving as the control. Then confirmed *Arabidopsis* OE lines were used for phenotype observation.

### 4.3. BjuAGL9-2 Interaction Protein Screening

The Matchmaker Gold Yeast two Hybrid System (Clontech, Tokyo, Japan) was used to screen proteins interacting with BjuAGL9-2. The bait plasmids pGBKT7-BjuAGL9-2 was co-transformed with a *B. juncea* bolting-stage flower buds yeast cDNA library (10^8^ cfu/mL) into *Saccharomyces cerevisiae* Y2H Gold competent cells. Transforms were selected on SD/-His-Leu-Trp medium plates, and single colonies were subsequently transferred into liquid QDO (SD/-Ade/-His-Leu-Trp) medium for further selection. PCR amplification was performed on positive clones, and the resulting products containing candidate interaction protein-encoding genes were sequenced. The sequencing results were analyzed using BLAST (Braju tum V 2.0 cds) searches against the NCBI database and the *Brassica* database (http://brassicadb.cn/#/GeneSequence/, accessed in 2 November 2025) to identify potential interactors.

### 4.4. The Interaction Analysis Between BjuAGL9-2 and Screened Proteins

To confirm the interactions between BjuAGL9-2 and candidate protein identified from the library screen, the bait construct pGBKT7-BjuAGL9-2 and prey constructs (pGADT7-BjuANXAD1, pGADT7-BjuTUA5, pGADT7-BjuZFP7, pGADT7-BjuGSTU5, pGADT7-BjuVA09G18110, pGADT7-BjuWRKY11, pGADT7-BjuRHY1A, pGADT7-BjuVA01G03870, and pGADT7-BjuMAPK 16) were co-transformed pairwise into the *Saccharomyces cerevisiae* Y2H Gold strain. Self-activation of BjuAGL9-2 was tested on SD/-Trp medium, and bait toxicity as well as recombinant plasmid activation were examined on SD/-Leu-Trp medium plates at 30 °C. Negative controls consisted of Y2H Gold [pGADT7-control] × [pGBKT7-lam], while positive controls were Y2H Gold [pGADT7-control] × [pGBKT7-p53]. Protein–protein interaction assays were performed on DDO (SD/-Leu-Trp) and QDO (SD/-Ade/-His-Leu-Trp/ABA/X-α-gal) medium plates. The plates were incubated at 30 °C for 3–5 days before analysis.

### 4.5. Bimolecular Florescence (BiFC) Analysis and Confocal Microscopy

BiFC assays were performed to further confirm the interaction between BjuAGL9-2 and candidate proteins. Briefly, the coding sequence of *BjuTUA5*, *BjuZFP7*, *BjuGSTU5*, and *BjuMAPK16* were cloned into the pEARLEYGATE201-YC vector, while *BjuAGL9-2* was cloned into the pEARLEYGATE202-YN vector via LR recombination. Each construct was introduced into *Agrobacterium tumefaciens* strain GV3101. Agrobacterium cultures carrying the respective constructs were co-infiltrated into leaves of 4-week-old *Nicotiana benthamiana*, together with the silencing suppressor P19. After 48–60 h, yellow fluorescent protein (YFP) signals were detected using a confocal laser scanning inverted microscope (LSM510 Meta; Carl Zeiss, Jena, Germany).

### 4.6. Sub-Cellular Localization Assays

The sub-cellular location of BjuAGL9-2 and its interacting proteins was examined using GFP fusion constructs, in which GFP was fused to BjuAGL9-2 and its interacting protein N-terminal. Briefly, the coding sequences of *BjuAGL9-2*, *BjuTUA5*, *BjuZFP7*, *BjuGSTU5*, and *BjuMAPK16* were cloned into the p1300-GFP vector. Each construct was transformed into *Agrobacterium tumefaciens* strain GV3101. Agrobacterium cultures carrying the constructs were co-infiltrated with the silencing suppressor P19 into leaves of 4-week-old *Nicotiana benthamiana* plants. After 48-60 h, green fluorescent protein (GFP) signals were observed using a fluorescence microscope.

## Figures and Tables

**Figure 2 plants-14-03502-f002:**
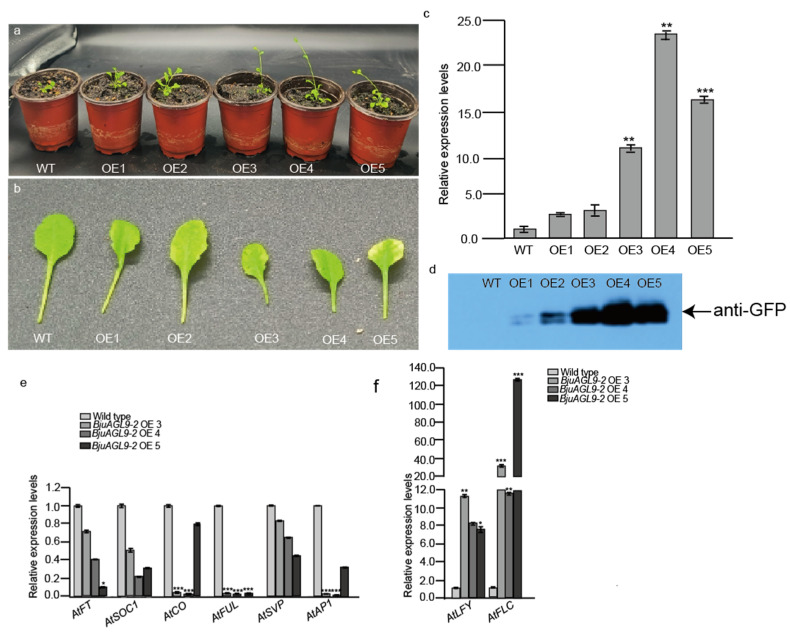
The phenotype observation of *Arabidopsis* OE lines. (**a**) The phenotype observation of *Arabidopsis* OE lines after transfer to nutrient soil after 28 days. (**b**) The third rosette leaf phenotype of *Arabidopsis* OE lines after transfer to nutrient soil after 28 days. (**c**) The mRNA expression levels of *BjuAGL9-2* in OE lines. (**d**) The protein expression level in OE lines. (**e**,**f**) The mRNA expression levels of floral integrator genes in *Arabidopsis* OE lines. WT: wild type, OE1-OE5: *Arabidopsis* over-expression line 1 to line 5. * *p* < 0.05, ** *p* < 0.01, *** *p* < 0.001.

**Figure 3 plants-14-03502-f003:**
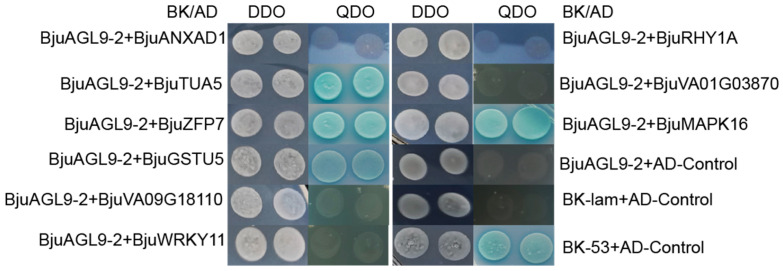
Interaction analysis between BjuAGL9-2 and screened proteins. Yeast strain Y2H Gold cells were co-transformed with the bait and prey transform plasmids and plated on SD/-Ade/-His-Leu-Trp/ABA/X-α-gal media.

**Figure 4 plants-14-03502-f004:**
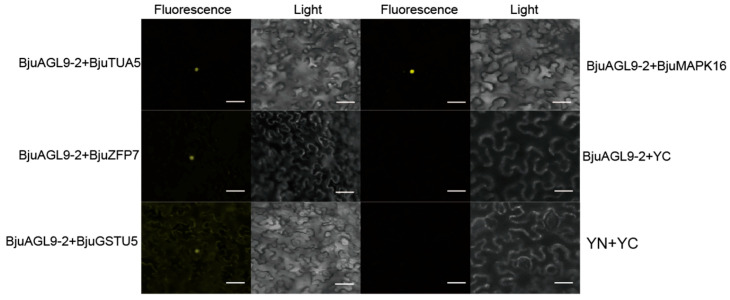
Bimolecular fluorescence complementation (BiFC) assays in tobacco leaf cells testing protein–protein interaction between BjuAGL9-2 with BjuTUA5, BjuZFP7, BjuGSTU5, and BjuMAPK16. Fluorescence, yellow fluorescence signal; light, bright light field; bar = 100 µm.

**Figure 5 plants-14-03502-f005:**
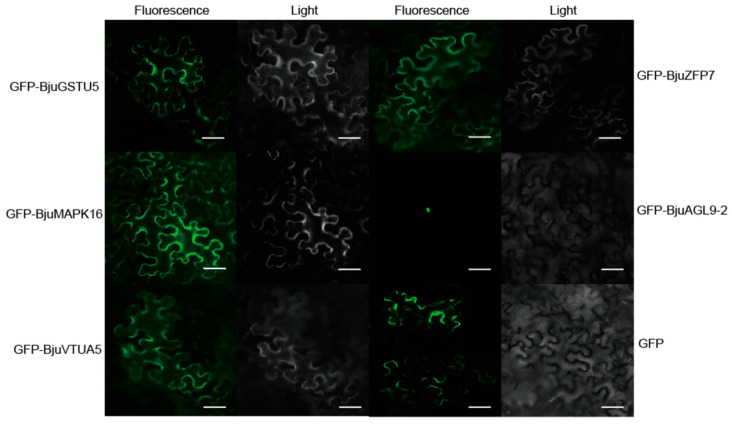
BjuAGL9-2, BjuTUA5, BjuZFP7, BjuGSTU5, and BjuMAPK16 sub-cellular location assays. Light: the image under bright light field; fluorescence: the image under GFP fluorescence. Bar = 100 µm.

**Figure 6 plants-14-03502-f006:**
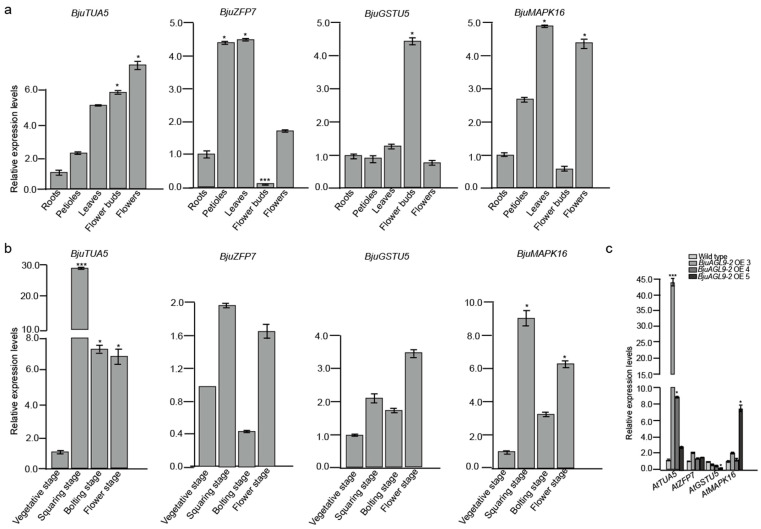
Transcription levels of genes producing interactive proteins in different tissues and at different developmental stages. (**a**) Transcription levels of genes producing interactive proteins in different tissues. (**b**) Interactive proteins encoding gene transcription levels at different developmental stages. (**c**) Transcription levels of homologs of interactive protein-encoding genes. OE3-OE5: *Arabidopsis* over-expression line 3 to line 5. * *p* < 0.05, *** *p* < 0.001.

**Figure 7 plants-14-03502-f007:**
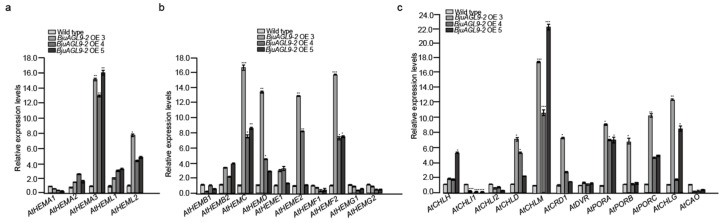
Chlorophyll synthesis gene expression levels in *Arabidopsis* OE lines. (**a**) The first step of chlorophyll synthesis gene expression levels in *Arabidopsis* OE lines. (**b**) The second step of chlorophyll synthesis gene expression levels in *Arabidopsis* OE lines. (**c**) The third step of chlorophyll synthesis gene expression levels in *Arabidopsis* OE lines. OE3-OE5: *Arabidopsis* over-expression line 3 to line 5. * *p* < 0.05, ** *p* < 0.01, *** *p* < 0.001.

**Table 1 plants-14-03502-t001:** Sequencing results of BjuAGL9-2 potential interacting proteins.

Gene ID	Gene Names	Similarity	Function
*BjuVA05G23270*	Annexin D1	100%	Abiotic stress, seedling development
*BjuVA10G20230*	Tubulin alpha-5 chain (TUA5)	99.19%	Pollen development
*BjuVA08G27100*	Zinc finger protein 7 (ZFP7)	97.96%	Trichome development
*BjuVA03G25670*	Glutathione S-transferase U5 (GSTU5)	100%	Defense response
*BjuVA09G18110*	Uncharacterized protein	100%	Unknown
*BjuVA03G59720*	WRKY transcription factor 11	98.17%	Abiotic stress responses
*BjuVA08G03470*	E3 ubiquitin-protein ligase RHY1A	100%	Protein degradation
*BjuVA01G03870*	Uncharacterized protein	93.42%	Unknown
*BjuVA10G20860*	Mitogen-activated protein kinase 16 (MAPK16)	100%	Defense response

## Data Availability

The original contributions presented in this study are included in the article/Appendix A. Further inquiries can be directed to the corresponding authors.

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
