# Peer review of "Ectopic Over-Expression of BjuAGL9-2 Promotes Flowering and Pale-Yellow Phenotype in Arabidopsis"

_plants, 2025, doi:10.3390/plants14223502_

Round 1

Reviewer 1 Report

Comments and Suggestions for Authors

Major Comments

  1. Although the study confirms that BjuAGL9-2 interacts with BjuTUA5, BjuZFP7, BjuGSTU5, and BjuMAPK16, and regulates the expression of chlorophyll biosynthesis genes, it fails to establish a complete evidence chain linking "gene-protein interaction-downstream pathway-phenotype".

  The mediating role of interacting proteins in flowering remains unvalidated. The study does not detect the expression changes of core floral integrator genes in Arabidopsis (e.g., FT, SOC1, LFY) after interaction, making it impossible to confirm whether BjuAGL9-2 regulates known flowering pathways through these interacting proteins. Therefore, I suggest that the authors to check the expression of core floral integrator genes in BjuAGL9-2 over-expression lines.

2.Limitations of experimental materials undermine the applicability of conclusions.

The study only verifies the function of BjuAGL9-2 through heterologous overexpression in Arabidopsis, lacking critical experimental support in its native host, Brassica juncea. BjuAGL9-2 knockout/knockdown lines in B. juncea are not constructed, nor is the phenotype of overexpressed Arabidopsis plants complemented in B. juncea. This makes it impossible to rule out species-specific interference from heterologous expression in Arabidopsis (e.g., the potential involvement of species-specific regulatory factors in B. juncea that are absent in the experiment). Therefore, I suggest that the authors to modify the conclusions and discussion about this section.

  1. Inadequate systematic analysis of chlorophyll biosynthesis regulation

Although the expression of 27 chlorophyll biosynthesis-related genes is detected, the lack of key data reduces the credibility of the association between phenotype and gene regulation. Lack of direct quantification of chlorophyll content. The "pale-yellow" phenotype is only described subjectively. Quantitative measurements (e.g., SPAD value, spectrophotometry) of chlorophyll a/b content and total chlorophyll concentration in OE lines are not conducted, making it impossible to quantify the actual impact of gene expression changes on chlorophyll biosynthesis. I suggest that the authors should measure chlorophyll a/b content and total chlorophyll concentration in OE lines.  

Meanwhile, contradictory data needs to be corrected. Section 2.6 clearly states "18 out of 27 genes were upregulated and 9 were downregulated," while the Discussion mentions "18 upregulated and 8 downregulated." The inconsistent data and unaddressed reasons affect the rigor of gene expression analysis.

  1. Uninvestigated interaction mechanism

The key interaction domains between BjuAGL9-2 and its interacting proteins (e.g., whether the MADS-box domain is involved in binding) are not analyzed. The molecular details of the interaction remain unclear. I suggest the authors use Y2H and BiFC to make sure which domains are key motifs between BjuAGL9-2 and its interacting proteins.

Minor comments

  1. Data Contradictions and Typos in Result Descriptions

There is a contradiction in the number of upregulated/downregulated chlorophyll biosynthesis genes (Section 2.6: "18 up, 9 down" vs. Discussion: "18 up, 8 down"). Original data should be checked and descriptions standardized.

Section 2.5 mentions "except AtGATU5." Combined with the interacting protein BjuGSTU5, its homologous gene in Arabidopsis should be AtGSTU5. "AtGATU5" is an obvious typo and needs correction to ensure consistency in gene nomenclature.

  1. Unclear Figure Labeling and Explanations

Subfigures (a) and (b) in Figure 2 are only labeled "Phenotype observation" and "Leaf phenotype," with no key information such as imaging stage (e.g., seedling age) or number of plants specified;

The p-value labels in Figure 6c "*: p<0.05, **: p<0.01, ***: p<0.001")are garbled or formatted incorrectly and should be standardized to “*p<0.05, **p<0.01, ***p<0.001”. The authors should check the whole manuscript to avoid the similar mistakes.

"Bar = 100 µm" is repeatedly labeled in Figure 5, and the format should be simplified.

  1. Missing Details in Methods and Insufficient Reproducibility

The source of the yeast two-hybrid library (e.g., specific tissue or developmental stage of B. juncea) and library titer are not specified, making it impossible to replicate the library construction process.

Only the "Agrobacterium-mediated floral dip method" is mentioned for Arabidopsis transformation, with no indication of whether the Agrobacterium strain (e.g., GV3101) carries a helper plasmid (e.g., pSoup).

For qRT-PCR, the primer design principles (e.g., whether primers span introns) and amplification efficiency verification results are not described.

The authors should check the whole manuscript and supplement the information.

  1. Lack of Quantification and Replication Data in Phenotype Observation

The early flowering phenotype is only described as "OE lines flowered at ~28 days, wild type at ~35 days," with no specification of the number of replicate plants per OE line (e.g., n=3 or n=5) or standard deviation, making it impossible to judge data reliability.

The "pale-yellow" leaf color lacks objective quantitative indicators (e.g., SPAD value) and relies on subjective judgment. Chlorophyll content measurements at different developmental stages should be supplemented.

  1. Inadequate Control Setup for Subcellular Localization Experiments

Only a GFP empty vector control is set up; no nuclear localization marker (e.g., H2B-RFP) is introduced to confirm the nuclear location, making it impossible to rule out errors in judging "BjuAGL9-2 localization in the nucleus". The authors should add the nuclear localization marker.

Parameters such as excitation wavelength and exposure time for fluorescence observation are not specified, resulting in opaque experimental conditions. The authors should check the whole manuscript and add the details in “Materials and Methods” section.

  1. Unclear Definition of Developmental Stages

Morphological criteria for defining "vegetative stage," "squaring stage," and "bolting stage" are not provided (e.g., "vegetative stage = 6 true leaves," "squaring stage = first flower bud diameter 2 mm"). Sample collection cannot be replicated in different laboratories, and morphological basis for stage division should be supplemented.

  1. Unspecified Screening Criteria for OE Lines

The methods only mention "5 BjuAGL9-2 OE lines were generated," with no specification of the number of initially screened positive clones or qRT-PCR screening thresholds (e.g., OE lines defined as those with expression levels >5-fold higher than wild type). The screening process lacks transparency.

  1. Undetected Expression of Key Flowering-Related Genes

The core of the study is "BjuAGL9-2 promotes flowering," but the expression changes of core floral integrator genes (FT, SOC1, LFY) in Arabidopsis are not measured, making it impossible to establish the association between BjuAGL9-2 and known flowering pathways. qRT-PCR data for these genes should be supplemented.

  1. Disorganized Reference Format

The format of author names in the reference list is inconsistent (e.g., a colon instead of a comma in "Fornara: F., De, M.A.")

  1. Missing Temporal Dynamic Observation of Leaf Color Phenotype

Only the pale-yellow phenotype of OE lines is described. Changes in this phenotype during the seedling, adult, and reproductive stages are not observed, making it impossible to judge whether leaf color changes are associated with developmental stages. If possibe, the authors shoud add the temporal series of phenotype photos and chlorophyll content data should be supplemented.

  1. Unclear Figures

  The column of BjuTUA5 is not well-presented. The authors should check the whole manuscript and solve all similar questions.

Author Response

Response to reviewers

Dear editors,

Thank you very much for your comments and suggestions with regard to our manuscript “Ectopic overexpression BjuAGL9-2 promoting flowering and pale-yellow in Arabidopsis”. These comments are valuable and helpful for revising and improving our paper. We have read the comments carefully and made revisions which highlighted in red in the manuscript according to your comments. The point-by-point answers to your comments were listed as below.

Comment 1: "I suggest that the authors to check the expression of core floral integrator genes in BjuAGL9-2 over-expression linesResponse: Thank you for pointing to check core floral integrator genes in BjuAGL9-2 overexpression lines. We already checked AtFT, AtSOC1, AtCO, AtFUL, AtSVP, AtLFY, AtLC and AtAP1 mRNA expression levels in BjuAGL9-2 over-expression lines. The results already marked in manuscript in red.

Comment 2: Limitations of experimental materials undermine the applicability of conclusions.

Response: Thank you for pointing to limitation of experimental materials. We already carefully checked and revised. The revised part was marked in red.

Comment 3: Modify the conclusions and discussion.

Response: Thank you for pointing to modify the conclusions and discussion. We already carefully checked and revised. The revised part was marked in red.

Comment 4: Suggest authors should measure chlorophyll a/b content and total chlorophyll concentration in OE lines

Response: Thank you for pointing to detect chlorophyll a/b content and total chlorophyll concentration in OE lines. Firstly, we designed the experiments, but we could not harvest enough leaves to do it. And we already observed obvious pale-yellow leaves. We think there is no doubt that the total chlorophyll concentration was decreased.

Comment 5: contradictory data needs to be corrected. Section 2.6 clearly states "18 out of 27 genes were up-regulated and 9 were down-regulated," while the Discussion mentions "19 up-regulated and 8 down-regulated."

Response: Thank you for pointing the contradictory data in manuscript. We already carefully checked and revised. The revised part was marked in red.

Comment 6: The authors use Y2H and BiFC to make sure which domains are key motifs between BjuAGL9-2 and its interacting proteins.

Response: Thank you for pointing to check the key motif between BjuAGL9-2 and its interacting proteins. And as we all know the protein structure is essential for protein interaction. In this study, we already checked the interaction between BjuAGL9-2 and screening proteins. We think it already enough to confirm the protein-protein interactions. And the next step, we will check key motifs between BjuAGL9-2 and its interacting proteins.

Comment 7: There is a contradiction in the number of up-regulated/down-regulated chlorophyll biosynthesis genes (Section 2.6: "18 up, 9 down" vs. Discussion: "18 up, 8 down"). Original data should be checked and descriptions standardized

Response: Thank you for pointing the contradiction in the number of up-regulated/down-regulated chlorophyll biosynthesis genes. We already carefully checked and revised. The revised part was marked in red.

Comment 8: Section 2.5 mentions "except AtGATU5." Combined with the interacting protein BjuGSTU5, its homologous gene in Arabidopsis should be AtGSTU5. "AtGATU5" is an obvious typo and needs correction to ensure consistency in gene nomenclature.

Response: Thank you for pointing the obvious typo of AtGSTU5. We already carefully checked and revised. The revised part was marked in red.

Comment 9: Subfigures (a) and (b) in Figure 2 are only labeled "Phenotype observation" and "Leaf phenotype," with no key information such as imaging stage (e.g., seedling age) or number of plants specified

Response: Thank you for pointing to describe plant growth stage in figure 2. We already carefully checked and revised. The revised part was marked in red.

Comment 10: The p-value labels in Figure 6c "*: p<0.05, **: p<0.01, ***: p<0.001") are garbled or formatted incorrectly and should be standardized to “*p<0.05, **p<0.01, ***p<0.001”.

Response: Thank you for pointing to incorrectly writing of p-value. We already carefully checked and revised. The revised part was marked in red.

Comment 11: Bar = 100 µm" is repeatedly labeled in Figure 5, and the format should be simplified.

Response: Thank you for pointing to repeat labeled Bar in figure 5. We already carefully checked and revised. The revised part was marked in red.

Comment 12: The source of the yeast two-hybrid library (e.g., specific tissue or developmental stage of B. juncea) and library titer are not specified, making it impossible to replicate the library construction process.

Response: Thank you for pointing to the source of yeast two-hybrid library. We already carefully checked and revised. The revised part was marked in red.

Comment 13: Only the "Agrobacterium-mediated floral dip method" is mentioned for Arabidopsis transformation, with no indication of whether the Agrobacterium strain (e.g., GV3101) carries a helper plasmid (e.g., pSoup)

Response: Thank you for pointing to much more describe the GV3101 strain. We bought GV3101 from shanghaiweidi. It contains a pJIC SA_Rep plasmid. And we already carefully checked and revised. The revised part was marked in red.   

Comment 14: For qRT-PCR, the primer design principles (e.g., whether primers span introns) and amplification efficiency verification results are not described.

Response: Thank you for your comment. We only used encoding sequence to design primers for qRT-PCR. And we only used a single peak primers set for qRT-PCR assays.

Comment 15: The early flowering phenotype is only described as "OE lines flowered at ~28 days, wild type at ~35 days," with no specification of the number of replicate plants per OE line (e.g., n=3 or n=5) or standard deviation, making it impossible to judge data reliability.

Response: Thank you for pointing to specification number of replicate plants for per OE lines. We already carefully checked and revised. The revised part was marked in red.   

Comment 16: The "pale-yellow" leaf color lacks objective quantitative indicators (e.g., SPAD value) and relies on subjective judgment. Chlorophyll content measurements at different developmental stages should be supplemented.

Response: Thank you for pointing to detect chlorophyll a/b content and total chlorophyll concentration in OE lines. Firstly, we designed the experiments, but we could not harvest enough leaves to do it. And we already observed obvious pale-yellow leaves. We think there is no doubt that the total chlorophyll concentration was decreased.

Comment 17: Only a GFP empty vector control is set up; no nuclear localization marker (e.g., H2B-RFP) is introduced to confirm the nuclear location, making it impossible to rule out errors in judging "BjuAGL9-2 localization in the nucleus". The authors should add the nuclear localization marker.

Response: Thank you for pointing to add a nuclear localization marker. Firstly, we design a nuclear localization marker, but unfortunately, our nuclear localization marker not working. We speculate the bacteria died. And we observed an obvious signal in BiFC assays. So we think it enough to confirm a nuclear location.

Comment 18: Parameters such as excitation wavelength and exposure time for fluorescence observation are not specified, resulting in opaque experimental conditions. The authors should check the whole manuscript and add the details in “Materials and Methods” section.

Response: Thank you for pointing to describe more detail for plant growth condition. We already carefully checked and revised. The revised part was marked in red.   

Comment 19: Morphological criteria for defining "vegetative stage," "squaring stage," and "bolting stage" are not provided

Response: Thank you for pointing to describe more detail for plant growth stages. We already carefully checked and revised. The revised part was marked in red. 

Comment 20: The methods only mention "5 BjuAGL9-2 OE lines were generated," with no specification of the number of initially screened positive clones or qRT-PCR screening thresholds (e.g., OE lines defined as those with expression levels >5-fold higher than wild type). The screening process lacks transparency

Response: Thank you for your comment to point the total number of BjuAGL9-2 OE lines. We already carefully checked and revised. The revised part was marked in red. At T0 generation, We get 21 BjuAGL9-2 OE lines; At T1 generation,We get 15 BjuAGL9-2 OE lines; At T2 generation, We get 8 BjuAGL9-2 OE lines, but we only harvest 5 lines seeds for T3 generation. The other 3 lines displayed much more serious pale-yellowing phenotype, and did not growth to reproductive stages.

Comment 21: The core of the study is "BjuAGL9-2 promotes flowering," but the expression changes of core floral integrator genes (FT, SOC1, LFY) in Arabidopsis are not measured, making it impossible to establish the association between BjuAGL9-2 and known flowering pathways. qRT-PCR data for these genes should be supplemented

Response: Thank you for pointing to check core floral integrator genes in BjuAGL9-2 overexpression lines. We already checked AtFT, AtSOC1, AtCO, AtFUL, AtSVP, AtLFY, AtLC and AtAP1 mRNA expression levels in BjuAGL9-2 over-expression lines. The results already marked in manuscript in red.

Comment 22: Disorganized Reference Format

Response: Thank you for pointing to disorganized reference format. We already carefully checked and revised. The revised part was marked in red.  

Comment 23: Missing temporal dynamic observation of leaf color phenotype

Response: Thank you for pointing to missing temporal dynamic observation of leaf color phenotype. We already carefully checked and revised. The revised part was marked in red.

Comment 24: The column of BjuTUA5 is not well-presented. The authors should check the whole manuscript and solve all similar questions.

Response: Thank you for pointing to not well-presented of BjuTUA5. We already carefully checked and revised. The revised part was marked in red.

Reviewer 2 Report

Comments and Suggestions for Authors

The manuscript submitted by Han et al. is interesting and contains many results. However, it needs improvement to demonstrate the value of the work performed.

In my opinion, the Title should be "Ectopic Overexpression of BjuAGL9-2 Promotes Flowering and Pale-Yellow Leaves in Arabidopsis" because a definition of what became pale yellow is necessary.

In the Abstract and the main text, the authors mention that the BjuAGL9-2 gene was identified from RNA-Seq data. However, they provide little information about this RNA-Seq (Lines 100-102: "Differentially expressed gene (DEG) analysis of shoot tip samples collected from both vegetative and reproductive stages identified BjuAGL9-2 as a gene significantly up-regulated during the reproductive stage.”). Have these results been published in any paper? Have the results been deposited in any database? Please provide the reference and complete information. How was this gene named? Was the name AGL9 chosen due to the protein similarity to another previously published AGL9? Why did they name the gene AGL9-2? Is there an AGL9-1? How many AGLs are there in the Brassica juncea genome?

When presenting the results, the authors should make clear that they overexpressed BjuAGL9-2 in Arabidopsis thaliana and NOT in B. juncea. This information appears only in the Title and in the Materials and Methods section. Additionally, it is essential to mention throughout the text that BjuAGL9-2 was overexpressed in fusion with GFP. It is also important to inform whether the fusion was in the N- or C-terminal of BjuAGL9-2, since a fusion protein may have its structure, cellular localization and/or function disturbed.

The sentence "Arabidopsis thaliana contains 108 MADS-box members, while Brassica rapa, Zea mays, and Vitis vinifera harbor 160, 87, and 54 members, respectively [10-13]." appears twice. It is first on lines 49-51 and again on lines 65-67.

In the Introduction, the authors describe the MADS-box genes and the ABCDE model. However, very little is revealed about the AGLs and, more specifically, what is known about AGL9 in other species.

Authors could include representative photos of the collected materials as supplemental figures, clarifying what was considered as vegetative, squaring, bolting, and flowering stages in this species.

In Figure 1b, which tissues were collected at the mentioned stages? The whole plant? Only the aerial part? Only the shoot tip?

In the legend for Figure 1, please include information about the meaning of the asterisks and the bars in the columns (standard deviation?). What statistical test was used? What was the number of biological replicates, and what were the expressions compared to for significance? For example, in Figure 1a, were the comparisons made with the expression in roots? Information needs to be written clearly for readers.

Figure 2a: What generation of transgenics is being analyzed? Are the plants homozygous? Are they the same age?

Figure 2b: Please, describe which leaf is being shown. The leaves are very different in size.

Figure 2c: What was considered as the wt expression level, since Arabidopsis does not have the BjuAGL9-2 gene?

Lines 129-130: The authors mention the construction of a cDNA library in the Y2H vector. However, they do not say which tissues were used for RNA extraction. Nor do they mention how many theoretical clones were analyzed in the screening. The authors report finding 12 clones, mention the insert sizes, and show a photo of a gel (Figure S1). However, they only describe the identity of 9 clones (Table 1). Were the others duplicate clones?

Table 1 shows the percentage of similarity. Similar to what?

Lines 149-150: It is written "… plated on SD/-Ade/-His-Leu-Trp/ABA/X-α-gal mediums." What does ABA mean? Aureobasidin? The authors do not mention anything about resistance to aureobasidin…

Figure 4 shows the BiFC results. They showed only one cell from each interaction, and the images are not very clear. The negative control for the other configuration of each interaction is missing. Additionally, the authors need to clarify whether the fusions are at the N or C terminus of the proteins under study.

Figure 5 shows the subcellular localization results. Here, too, the images are not very clear. Apparently, the GFP fusions were made at the N-terminus of all proteins. All proteins that are candidates for interaction with BjuAGL9-2 are in the cytoplasm. How do the authors explain that the interaction occurs in the nucleus? This point requires further discussion. Is it possible that the GFP fusion is masking their nuclear localization signals?

Figure 6b: In the graphic of BjuTUA5, part of the column "squaring stage" is missing.

In the legend of Figure 6, please clearly indicate for each item which species was used for RNA extraction. Additionally, the authors should explain the criteria used to define homologous sequences in Arabidopsis. Did they produce phylogenetic trees with the sequences of the proteins under study in both species?

Lines 203-205: It is written "… qRT-PCR assays indicated that except AtGSTU5 was down-regulated, AtTUA5, AtZFP7 and AtMAPK16 were up-regulated in BjuAGL9-2 OE lines, compared to wild type (Figure 6c)." However, only AtTUA5 is upregulated in 2 out of 3 OE lines, whereas the other genes are upregulated in only 1 out of 3 lines.

The authors analyzed the expression of several genes involved in chlorophyll synthesis. Some genes are upregulated, while others are downregulated. Did the authors draw any conclusions from these experiments? How can the pale yellow leaves phenotype be correlated with the expression of these genes?

Lines 227-228: Please, correct to "According to the ABCDE MODEL of FLOWER development, …"

The discussion needs improvement to deepen the comparisons between the results obtained and what is known about AGL9 in the literature.

The Materials and Methods section also needs improvement, with more detailed descriptions of the experiments performed. Some examples: Line 346: "natural field conditions" – Please, inform the time of year and the weather conditions during the period in which samples were taken, such as average temperature and length of the days.; Line 351: Please, provide more information about how the cDNA synthesis was performed, since Table S1 only presents primer sequences.; Line 358: "wild type plants" - Since BjuAGL9-2 overexpression was done in fusion with GFP, the ideal is to compare with mock plants, overexpressing GFP alone.; Line 362: "qRT-PCR" Please, provide more information about the experiment, such as amount of RNA/cDNA used, which enzyme was used, and what was used to produce fluorescence? SYBRGreen?; Line 364: "was analyzed in triplicate." Please define the number of biological and technical replicates.; Lines 385-387: Self-activation must be tested in the same medium as the assay, without HIS and without ADE.

The manuscript would benefit from some English correction, as the reader's understanding was impaired in a few places. Examples: Lines 53-54 – "… conserved MADS-box domain together with a highly C-terminal region [14]." There is a word missing after highly.; Lines 185-186 – "BjuGSTU5 displayed little change in petioles, …" Please, clarify "little change" in relation to what?; Line 266: "AMONG the five genes tested, …"

Reviewer 3 Report

Comments and Suggestions for Authors

The manuscript text is devoted to the study of a gene that, when overexpressed, showed a connection with early flowering and a pale-yellow leaf phenotype. I have some comments on the text, which I will outline line by line:

68-70    It is better to present the list of classes in a table or diagram, as the text is overloaded.
82–84    “Transgenic plants over-expression BjuAGL9-2” — it seems that was meant "overexpressing".
99-122    I don't understand which plants are being discussed here: transformed Arabidopsis or Brassica?! Apparently, the latter... In any case, statistical data (p-values, n samples) are needed. The text should also be supplemented and improved so that the results can be read unambiguously!
119–121    “All five transgenic lines showed a pale-yellow phenotype” — add a quantitative assessment (e.g., chlorophyll content or any spectrophotometric assay).
190    Image is unreadable
207-213    Excessive detail in the text; better to put it in a table or appendix.
224–232    The beginning of the section duplicates the introduction.
345–349    No information is available about lighting conditions and temperature.
352     "over Expression" -> overexpression
363    The qRT-PCR method is described too superficially—there is no mention of the device model used or the amplification conditions.
417-548    Several links (e.g., 39 and 51) duplicate each other — check for repetition.

The introduction is overloaded with repetitive descriptions of MADS-box genes (pp. 47–67), which disrupts the flow and reduces the scientific density. Some results (e.g., protein localization and interaction) are actually duplicated in several sections and annotations. The discussion (pp. 224–343) partially repeats the introduction and description of the results, but does not interpret them in the context of broader publications. I would recommend condensing the theoretical overview in the introduction, leaving only the information necessary to understand the purpose of the work. Remove repeated retellings of the results; leave the interpretation and connections between flowering regulation and chlorophyll change. The logical connection between the interaction of BjuAGL9-2 and leaf color change is not sufficiently argued. It is unclear whether similar phenotypes are observed in Arabidopsis SEP3-overexpression lines—there is no comparison with the literature. The author concludes that there is “regulation of chlorophyllogenesis” without providing direct evidence (e.g., analysis of pigment content).

Round 2

Reviewer 1 Report

Comments and Suggestions for Authors

The authors have solved the comments in the new version. It can be accepted by after finishing the grammar errors.

Author Response

Response to reviewer 1 Dear editors,
Thank you very much for your comments and suggestions with regard to our manuscript “Ectopic overexpression of BjuAGL9-2 promotes flowering and pale-yellow in Arabidopsis”. These comments are valuable and helpful for revising and improving our paper. We have read the comments carefully and made revisions which highlighted in red in the manuscript according to your comments. The point-by-point answers to your comments were listed as below. Comment 1: The authors have solved the comments in the new version. It can be accepted by after finishing the grammar errors.
Response: Thank you for pointing to grammar errors. We already carefully checked and revised. The revised was marked in red.

Reviewer 2 Report

Comments and Suggestions for Authors

I read version 2 submitted by the authors. They made several of the requested modifications, but they also failed to make other important changes. They do not describe the RNA-Seq nor explain the motivation for choosing the gene studied. They failed to improve the microscopy images in Figures 4 and 5. Additionally, the authors present simple explanations, without discussing important issues such as protein interactions in depth. Perhaps I am being too harsh, but I think the manuscript is not yet ready for publication.

Author Response

Response to reviewer 2 Dear editors,
Thank you very much for your comments and suggestions with regard to our manuscript “Ectopic overexpression of BjuAGL9-2 promotes flowering and pale-yellow in Arabidopsis”. These comments are valuable and helpful for revising and improving our paper. We have read the comments carefully and made revisions which highlighted in red in the manuscript according to your comments. The point-by-point answers to your comments were listed as below. Comment 1: They do not describe the RNA-Seq nor explain the motivation for choosing the gene studied.
Response: Thank you for pointing to describe RNA-Seq nor explain the motivation for choosing the gene studied. We already carefully checked and revised. The revised part was marked in red. Comment 2: They failed to improve the microscopy images in Figures 4 and 5.
Response: Thank you for pointing to improve the microscopy image in Figure 4 and Figure 5. We already carefully checked and revised. We adjusted the brightness and darkness and the image much more clearer. Comment 3: The authors present simple explanations, without discussing important issues.
Response: Thank you for pointing to present more explanations. We already carefully checked and revised. The revised part was marked in red.

Reviewer 3 Report

Comments and Suggestions for Authors

Contradiction: Abstract says AtFT/SOC1 were up-regulated, text says down-regulated.

The authors have clearly made major improvements in the scientific structure, logical flow, and methodological clarity of the paper. However, the linguistic quality and data accuracy deteriorated due to numerous typos, syntax errors, and inconsistencies.

Comments on the Quality of English Language

line:

55 “it typically contain” → “it typically contains”

262 "It 262
means that the interacting proteins may were transported to nucleus for their interac- 263
tion" -- I didn't understand that

Before journal submission, a thorough English proofreading, data consistency check, and style correction are strongly recommended.

Author Response

Response to reviewer 3 Dear editors,
Thank you very much for your comments and suggestions with regard to our manuscript “Ectopic overexpression of BjuAGL9-2 promotes flowering and pale-yellow in Arabidopsis”. These comments are valuable and helpful for revising and improving our paper. We have read the comments carefully and made revisions which highlighted in red in the manuscript according to your comments. The point-by-point answers to your comments were listed as below. Comment 1: Contradiction: Abstract says AtFT/SOC1 were up-regulated, text says down-regulated.
Response: Thank you for pointing to contradiction in manuscript. We already carefully checked and revised. The revised part was marked in red. Comment 2: The linguistic quality and data accuracy deteriorated due to numerous typos, syntax errors, and inconsistencies.
Response: Thank you for pointing to the linguistic quality and data accuracy deteriorated due to numerous typos, syntax errors, and inconsistencies. We already carefully checked and revised. The revised part was marked in red.
